# Risk of metabolic syndrome in participants within the normal range of alanine aminotransferase: A population-based nationwide study

**Ju-Yeon Cho[1]☯, Jae Yoon Jeong[2]☯, Won Sohn [3]\***

1 Division of Gastroenterology, Department of Internal Medicine, Chosun University Hospital, Gwang-Ju, Republic of Korea, 2 Division of Gastroenterology, Department of Internal Medicine, National Medical Center, Seoul, Republic of Korea, 3 Division of Gastroenterology, Department of Internal Medicine, Kangbuk Samsung Hospital, Sungkyunkwan University School of Medicine, Seoul, Republic of Korea

☯ These authors contributed equally to this work.
* wonsohn1@gmail.com

## Abstract

This study aimed to investigate the risk of metabolic syndrome (MS) in participants whose alanine aminotransferase (ALT) levels were within the normal range in the general population. A cross-sectional study was conducted using nationally representative samples from the Korea National Health and Nutrition Examination Survey 2007–2015. A total of 43,402 adults (men, 17,535; women, 25,867) with ALT ≤40 U/L without a history of hepatitis B and C, liver cirrhosis, or liver cancer were analyzed. The risk of MS was evaluated according to the ALT level. The prevalence of MS significantly increased as the ALT levels increased. The proportions of MS in men were 12.6%, 25.2%, and 39.7% in the ALT levels of <15, 15~30, and 30~40 U/L, respectively ($p < 0.001$), and those of women were 7.2%, 23.3%, and 44.7% in the ALT levels of <10, 10~20, and 20~40 U/L, respectively ($p < 0.001$). There was an ALT-dependent relationship in the risk of MS in participants with normal ALT level after adjustment for age, alcohol intake, and body mass index. The adjusted odds ratio (aOR) of MS in men was 2.48 (95% confidence interval [CI], 2.16–2.85) in an ALT level of 30~40 U/L compared with that in ALT <15 U/L ($p < 0.001$), and the aOR of MS in women was 2.67 (95% CI, 2.26–3.15) in an ALT level of 20~40 U/L compared with that in ALT <10 U/L ($p < 0.001$). Although within the normal range of ALT, the risk of MS increases as the ALT levels increase. The ALT level in the general population without a history of chronic liver disease may be a useful marker to evaluate for MS.

## Introduction

Metabolic syndrome (MS) is a cluster of conditions that predispose type 2 diabetes mellitus and cardiovascular disease. The prevalence of MS has increased in recent years worldwide [1]. It is noted for its clinical consequences because it is regarded as a precursor of cardiovascular

**Data Availability Statement:** All relevant data are within the manuscript and its Supporting Information files.

**Funding:** This study was supported by Research fund from Chosun University Hospital, 2019. The funders had no role in study design, data collection and analysis, decision to publish, or preparation of the manuscript.

**Competing interests:** The authors have declared that no competing interests exist.

diseases such as coronary artery and cerebrovascular diseases and type 2 diabetes mellitus [2]. MS is an independent risk factor for various cardiovascular conditions such as microvascular dysfunction, coronary calcification and atherosclerosis, myocardial infarction, and heart failure [3].

The measurement of alanine aminotransferase (ALT) level is a fundamental test in screening for liver disease and assessing disease progression. Abnormal serum ALT levels are found in a variety of liver diseases such as viral hepatitis, alcoholic liver disease, or nonalcoholic fatty liver disease (NAFLD). The risk of cardiovascular disease such as ischemic heart disease or type 2 diabetes mellitus also increases in the case of elevated ALT levels [4, 5]. Furthermore, the risk of MS increases in participants with elevated ALT levels [6, 7]. The prevalence and development of MS are associated with the increased ALT levels independently of insulin resistance [8, 9].

Most clinicians carefully examine an abnormal ALT level, while they have little interest in an ALT level within the normal range. However, several studies were conducted to re-evaluate the upper limit of normal (ULN) of ALT and suggested that the new ULN of ALT should be lower than the conventional one (<40 U/L) [10–12]. Some studies suggested that the ULN of ALT in healthy participants are defined as 29–33 and 19–25 U/L in males and females, respectively [13–15]. Therefore, this study planned to investigate the risk of MS in participants with normal ALT levels in the Korean general population using a population-based nationwide data. Furthermore, we evaluated an ALT-dependent relationship in the risk of MS in participants with normal ALT levels.

## Methods

### Study population

This cross-sectional study was conducted using data from the Korea National Health and Nutrition Examination Survey (KNHANES) between 2007 and 2015. The KNHANES, a nationally representative survey, is performed by the Korea Centers for Disease Control and Prevention (KCDC). It is based on a complex, stratified, multistage, and probability cluster sampling of the noninstitutionalized population in Korea [16]. This survey consists of four parts: health interview, nutrition, health behavior, and health examination surveys. The health interview and examination are performed by trained medical staffs and interviewers at the mobile examination center. The KNHANES has been periodically performed since 1998. This study was conducted using KNHANES IV (2007–2009), V (2010–2012), and VI (2013–2015).

The target population of this survey was all noninstitutionalized Korean civilians older than 1 year of age. The survey was conducted with the sampling units based on gender, age, and geographic areas, which were determined according to the household registries of the Korea National Census Registry. Written informed consent was obtained from all participants in the survey. The KNHANES was approved by the Institutional Review Board of KCDC (2007-02CON-04-P, 2008-04EXP-01-C, 2009-01CON-03-2C, 2010-02CON-21-C, 2011-02CON-06-C, 2012-01EXP-01-2C, 2013-07CON-03-4C, 2013-12EXP-03-5C, and 2015-01-02-6C).

A total of 73,353 participants completed the survey through the mobile health exam units. First, we excluded 24,499 participants according to the following criteria: age under 19 years old (n = 17,314), absence of laboratory data (n = 6,337), and absence of alcohol intake history (n = 848). Of the remaining 48,854 participants, we additionally excluded 2,026 patients with hepatitis B (n = 1,757), hepatitis C (n = 174), liver cirrhosis (n = 79), and liver cancer (n = 16). A total of 46,828 participants without a history of viral hepatitis, cirrhosis, or liver cancer were determined. Finally, 43,402 participants whose ALT levels were within the normal range (≤40

U/L) were included in the final analysis, and we compared them to the remaining 3,426 with ALT level >40 U/L (Fig 1).

## Clinical variables

The information about alcohol consumption was obtained during the health interview survey. Alcohol consumption was evaluated by questioning the participants about their drinking behavior during the month just before the interview. They were asked for their average frequency (days per month) of alcoholic beverage consumption and average amount (units of drink/day) of alcoholic drinks ingested on a single occasion. Each unit was equivalent to approximately 10 g of alcohol intake.

Blood tests including aspartate aminotransferase (AST); ALT; total, low-density lipoprotein (LDL), and high-density lipoprotein (HDL) cholesterol; and triglyceride were checked after 12 h of fasting. Routine biochemical tests, including triglyceride; glucose; total, HDL, and LDL cholesterol; ALT; and AST, were performed by ADVIA 1650 analyzer (Bayer, Pittsburgh, PA, USA). Hepatitis B surface antigen was measured using an electrochemiluminescence immunoassay method with an E-170 automated analyzer (Roche, Penzberg, Germany). Chemiluminescent microparticle immunoassay was performed to check the antibody to hepatitis C virus (anti-HCV) using ARCHITECT Anti-HCV (ABBOTT Diagnostics Division, Korea/Germany). HCV RNA was measured by real-time polymerase chain reaction using the COBAS AmpliPrep/COBAS Taq-Man HCV test (Roche, Penzberg, Germany). The hepatic steatosis index (HSI) was calculated to assess the relationship between the serum ALT levels and fatty liver grade [17]. High-risk alcohol consumption was defined as alcohol intake ≥7 drink-units one time and ≥2 times a week in men or ≥5 drink-units one time and ≥ 2 times a week in women. MS was defined according to the updated National Cholesterol Education Program Adult Treatment Panel III standards [18, 19].

## Statistical analysis

All analyses were performed based on gender. Categorical variables are presented as frequencies and percentages, whereas continuous variables are demonstrated as mean value with standard deviation and median value with interquartile range. The linear trend was analyzed between clinical variables and ALT levels categorized by 3 groups (male, ALT, 0–15, 15–30, and 30–40 IU/L; female, ALT, 0–10, 10–20, and 20–40 IU/L). The ALT cutoff level was defined according to the quartile distribution of ALT level (<25, 25–75, and ≥75 percentile). Continuous variables were analyzed by the weighted linear trend in the one-way ANOVA test, while categorical variables were analyzed by the linear-by-linear association in the chi-square test. A multivariable logistic regression model was used to investigate the influence of the ALT levels on MS. The risk of MS are presented as adjusted odds ratio (aOR). Three models were calculated according to each adjusting variable: age in model 1; age and alcohol consumption in model 2; age, alcohol consumption, and body mass index (BMI) in model 3. Multivariable analysis was performed using a forward conditional stepwise procedure to avoid multicollinearity. P values < 0.05 were considered statistically significant. All statistical analyses were performed using SPSS for Window release 18.0 (SPSS Inc., Chicago, IL, USA).

## Results

### Baseline characteristics of the participants whose ALT levels were within the normal range (≤40 U/L)

Table 1 presents the baseline characteristics of the participants whose ALT levels were within the normal range (≤40 U/L). The mean ages and BMI values of male and female participants

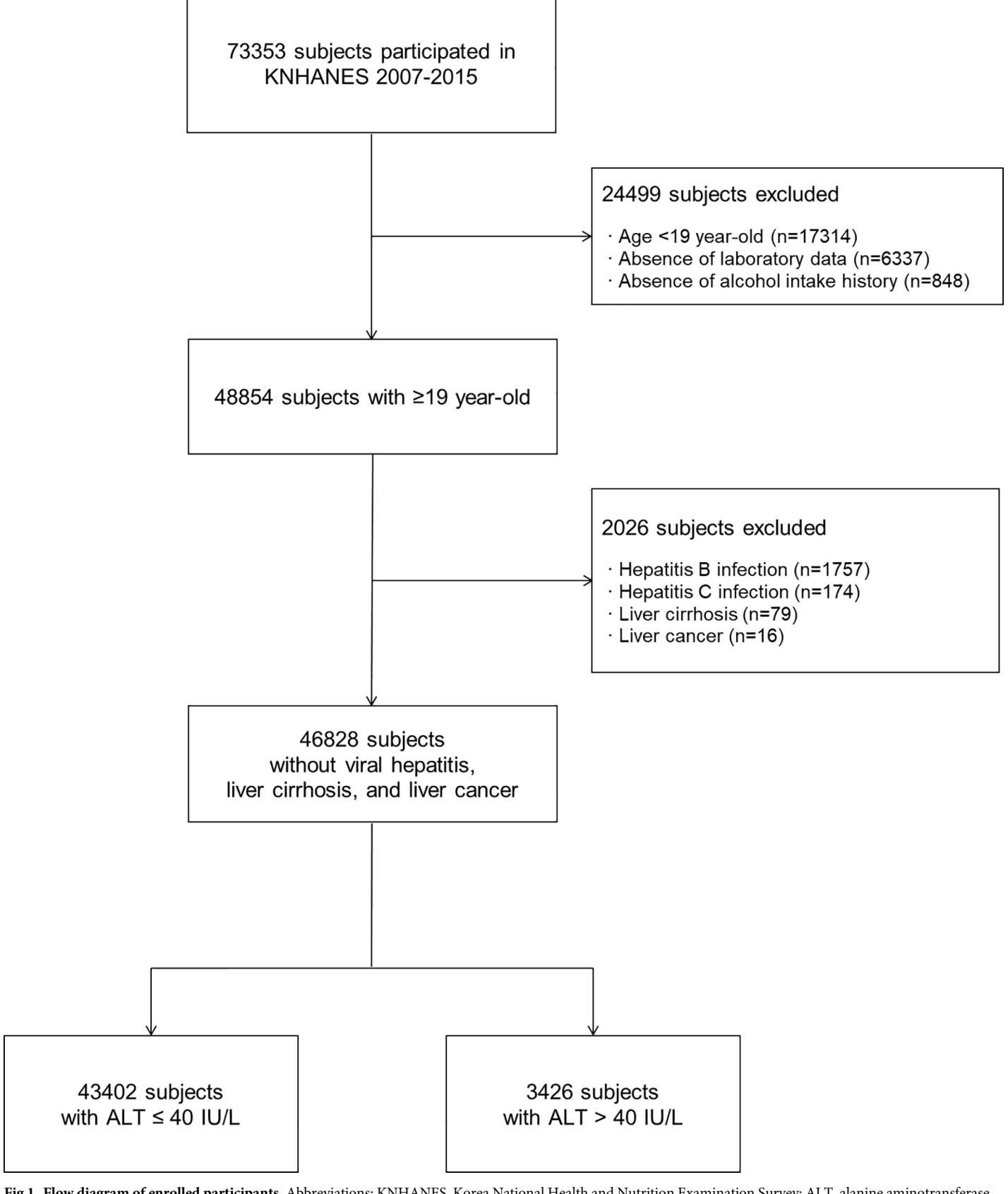

**Fig 1. Flow diagram of enrolled participants.** Abbreviations: KNHANES, Korea National Health and Nutrition Examination Survey; ALT, alanine aminotransferase.

Table 1. Baseline characteristics of the participants.

| | Male (n = 17,535) | Female (n = 25,867) |
|---|---|---|
| Age (year, mean ± SD) | 50.5 ± 16.6 | 49.5 ± 16.5 |
| Height (cm, mean ± SD) | 169.5 ± 6.7 | 156.6 ± 6.6 |
| Weight (kg, mean ± SD) | 68.2 ± 10.4 | 57.1 ± 8.8 |
| BMI (kg/m$^2$, mean ± SD) | 23.8 ± 3.0 | 23.3 ± 3.4 |
| Waist circumference (cm, mean ± SD) | 83.9 ± 8.6 | 78.5 ± 9.7 |
| SBP (mmHg, mean ± SD) | 121.2 ± 15.9 | 116.4 ± 18.0 |
| DBP (mmHg, mean ± SD) | 78.1 ± 10.6 | 73.6 ± 10.0 |
| AST (U/L, mean ± SD) | 22.0 ± 7.4 | 19.5 ± 5.8 |
| ALT (U/L, mean ± SD) | 20.8 ± 7.6 | 15.9 ± 6.5 |
| FBS (mg/dL, mean ± SD) | 100.2 ± 23.7 | 96.2 ± 21.2 |
| Total cholesterol (mg/dL, mean ± SD) | 185.6 ± 34.6 | 189.5 ± 36.2 |
| HDL cholesterol (mg/dL, mean ± SD) | 46.9 ± 11.0 | 52.0 ± 11.8 |
| Triglyceride (mg/dL, mean ± SD) | 148.8 ± 116.9 | 114.8 ± 77.4 |
| BUN (mg/dL, mean ± SD) | 15.4 ± 4.6 | 13.8 ± 4.4 |
| Creatinine (mg/dL, mean ± SD) | 1.0 ± 0.3 | 0.7 ± 0.2 |
| Hepatic steatosis index | 31.6 ± 4.5 | 32.0 ± 4.5 |
| High-risk alcohol consumption (N, mean ± SD) | 3313 (18.9%) | 1087 (4.2%) |
| Diabetes mellitus (N, %) | 1558 (8.9%) | 1708 (6.6%) |
| Use of antidiabetic agents | 1351 (7.7%) | 1543 (6.0%) |
| Use of lipid-lowering agents | 825 (4.7%) | 1768 (5.6%) |
| Use of antihypertensive agents | 3295 (18.8%) | 4872 (18.8%) |
| Metabolic syndrome (N, %) | 4312 (24.6%) | 6809 (26.3%) |
| Scores of metabolic syndrome 0/1/2/3/4/5 (N) | 4421/4626/4176/2734/1309/269 | 6873/6780/5405/3853/2270/686 |

Abbreviations: SD, standard deviation; BMI, body mass index; SBP, systolic blood pressure; DBP, diastolic blood pressure; AST, aspartate aminotransferase; ALT, alanine aminotransferase; FBS, fasting blood sugar; HDL, high-density lipoprotein; BUN, blood urea nitrogen.

were 50.5 and 49.5 years old and 23.8 and 23.3 kg/m$^2$, respectively. The systolic blood pressure (SBP) levels of male and female participants were 121.2 ± 15.9 and 116.4 ± 18.0, and their diastolic blood pressure (DBP) levels were 78.1 ± 10.6 and 73.6 ± 10.0, respectively. The ALT level of male participants was 20.8 ± 7.6 U/L, while that of females participants was 15.9 ± 6.5 U/L. The mean levels of the HSI in male and female participants were 31.6 ± 4.5 and 32.0 ± 4.5, respectively. The prevalence rates of MS in male and female participants were 24.6% and 26.3%, respectively. The proportion of high-risk alcohol consumption in male participants was 18.9%, while that in female participants was 4.2%.

Table 2 demonstrates the changes of clinical variables according to the ALT level. There were dose-dependent increases of BMI, waist circumference, SBP, DBP, fasting blood glucose (FBS), total cholesterol, triglyceride, and HSI and prevalence rates of diabetes mellitus and MS as the ALT levels increased in both male and female participants. In addition, the HDL cholesterol level decreased as the ALT levels increased in both groups. In male participants, the proportion of high-risk alcohol consumption increased as the ALT levels increased, whereas that in female participants had an inverse correlation with the increase in the ALT levels.

## Prevalence of MS in participants whose ALT levels were within the normal range

Fig 2 presents the prevalence of MS according to the ALT level. In male participants, the prevalence rates of MS were 12.6%, 25.2%, and 39.7% in the ALT levels of 0–15, 15–30, and 30–40

**Table 2. Clinical characteristics of the participants according to the ALT level.**

| | Male (n = 17,535) | | | | Female (N = 25,867) | | | |
|---|---|---|---|---|---|---|---|---|
| | 0~15 | 15~30 | 30~40 | *P* value for trend | 0~10 | 10~20 | 20~40 | *P* value for trend |
| Number of participants (No, %) | n = 3,927 | n = 10,916 | n = 2,692 | | n = 3,193 | n = 1,6601 | n = 6,073 | |
| Age (year, mean ± SD) | 50.2 ± 19.5 | 51.1 ± 15.9 | 48.4 ± 14.7 | <0.001 | 37.6 ± 15.1 | 49.9 ± 16.4 | 54.8 ± 14.2 | <0.001 |
| BMI (kg/m², mean ± SD) | 22.3 ± 2.7 | 23.9 ± 2.9 | 25.3 ± 3.0 | <0.001 | 21.4 ± 2.8 | 23.1 ± 3.2 | 24.9 ± 3.6 | <0.001 |
| Waist circumference (cm, mean ± SD) | 79.9 ± 8.3 | 84.4 ± 8.2 | 88.0 ± 8.1 | <0.001 | 72.9 ± 8.2 | 77.8 ± 9.2 | 83.2 ± 9.8 | <0.001 |
| SBP (mmHg, mean ± SD) | 119.0 ± 16.5 | 121.6 ± 15.7 | 122.8 ± 15.3 | <0.001 | 107.6 ± 14.5 | 116.2 ± 17.9 | 121.7 ± 18.0 | <0.001 |
| DBP (mmHg, mean ± SD) | 75.2 ± 10.3 | 78.5 ± 10.3 | 80.8 ± 10.8 | <0.001 | 69.8 ± 9.1 | 73.4 ± 9.9 | 76.1 ± 10.1 | <0.001 |
| AST (U/L, mean ± SD) | 17.7 ± 4.1 | 22.1 ± 6.0 | 28.3 ± 10.8 | <0.001 | 14.8 ± 5.4 | 18.4 ± 3.9 | 24.7 ± 6.6 | <0.001 |
| ALT (U/L, mean ± SD) | 11.8 ± 2.0 | 20.7 ± 4.1 | 34.1 ± 3.1 | <0.001 | 8.0 ± 1.2 | 14.0 ± 2.7 | 25.5 ± 5.1 | <0.001 |
| FBS (mg/dL, mean ± SD) | 97.1 ± 22.9 | 100.4 ± 23.2 | 103.6 ± 26.0 | <0.001 | 90.1 ± 15.0 | 95.2 ± 19.7 | 102.0 ± 26.2 | <0.001 |
| Total cholesterol (mg/dL, mean ± SD) | 176.6 ± 32.2 | 187.0 ± 34.1 | 192.9 ± 37.7 | <0.001 | 176.4 ± 32.7 | 189.3 ± 35.2 | 197.0 ± 38.5 | <0.001 |
| HDL cholesterol (mg/dL, mean ± SD) | 48.4 ± 11.2 | 46.7 ± 10.9 | 45.2 ± 10.9 | <0.001 | 54.9 ± 11.6 | 52.3 ± 11.8 | 49.6 ± 11.6 | <0.001 |
| Triglyceride (mg/dL, mean ± SD) | 112.0 ± 75.1 | 151.2 ± 117.6 | 192.6 ± 144.6 | <0.001 | 84.7 ± 48.8 | 109.7 ± 70.3 | 144.7 ± 96.6 | <0.001 |
| Hepatic steatosis index | 28.0 ± 3.2 | 31.9 ± 3.8 | 35.9 ± 4.3 | <0.001 | 28.0 ± 3.0 | 31.4 ± 3.8 | 35.7 ± 4.4 | <0.001 |
| High-risk alcohol consumption (N, mean ± SD) | 534 (13.6%) | 2121 (19.4%) | 658 (24.4%) | <0.001 | 165 (5.2%) | 705 (4.2%) | 217 (3.6%) | 0.001 |
| Diabetes mellitus (N, %) | 332 (8.5%) | 936 (8.6%) | 288 (10.7%) | <0.001 | 66 (2.1%) | 943 (5.7%) | 699 (11.5%) | <0.001 |
| Metabolic syndrome (N, %) | 495 (12.6%) | 2,748 (25.2%) | 1,069 (39.7%) | <0.001 | 231 (7.2%) | 3,862 (23.3%) | 2,716 (44.7%) | <0.001 |

Abbreviations: ALT, alanine aminotransferase; BMI, body mass index; SBP, systolic blood pressure; DBP, diastolic blood pressure; FBS, fasting blood sugar; HDL, high-density lipoprotein; HSI, hepatic steatosis index.

U/L, respectively. A dose-dependent relationship between MS and ALT levels was noted in male participants whose ALT levels were within the normal range ($p < 0.001$). In female

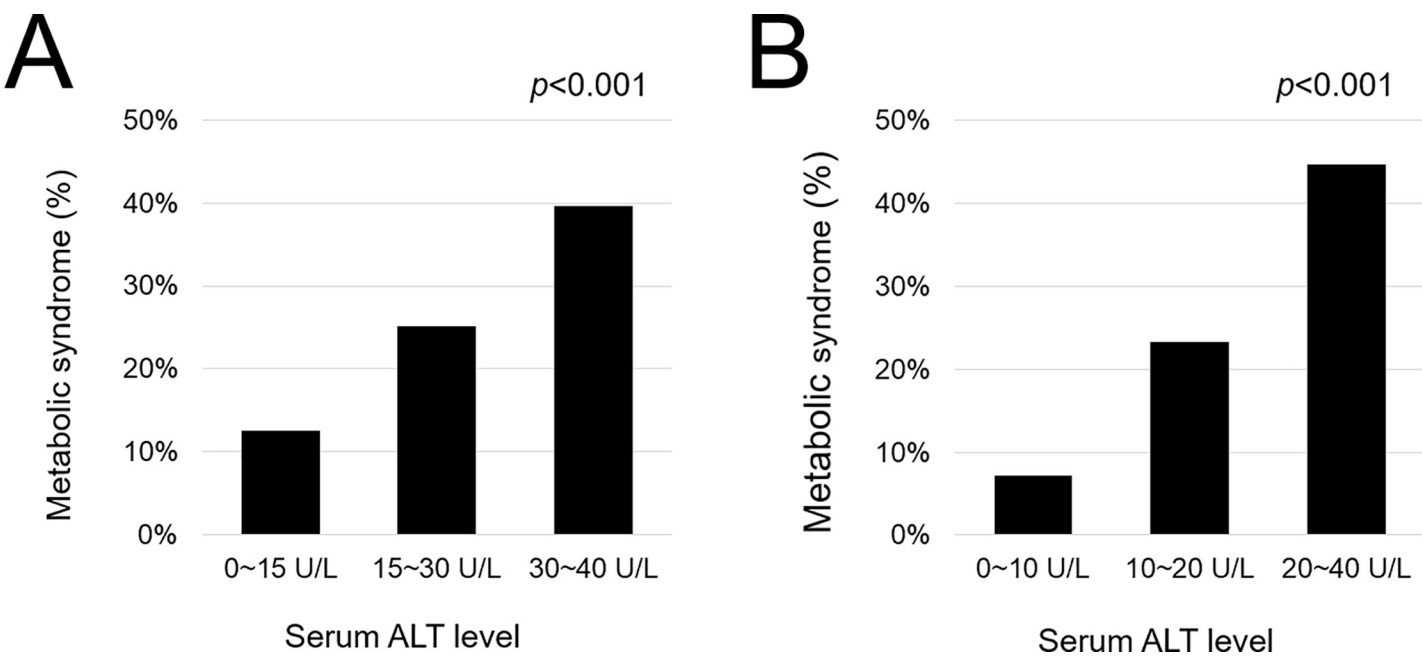

**Fig 2.** The prevalence of metabolic syndrome in patients with ALT ≤40 U/L: Male (A) and female (B). Abbreviation: ALT, alanine aminotransferase.

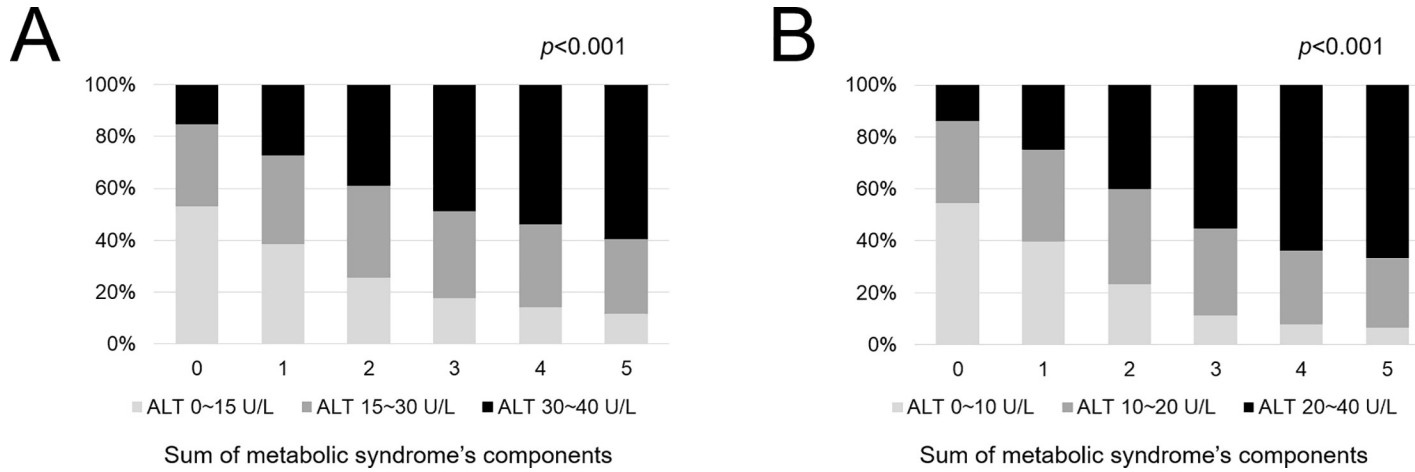

**Fig 3.** Differences of the ALT levels according to the sum of the metabolic syndrome scores: Male (A) and female (B). Abbreviation: ALT, alanine aminotransferase.

participants, the prevalence rates of MS were 7.2%, 23.3%, and 44.7% in the ALT levels of 0–10, 10–20, and 20–40 U/L, respectively. A dose-dependent relationship between MS and ALT levels was also noted in female participants whose ALT levels were within the normal range ($p < 0.001$). We also examined the relationship between the score of MS components and ALT levels (Fig 3). Consequently, a positive correlation in the increase of ALT levels and that of MS components was found in both male and female participants, respectively ($p < 0.001$ and $p < 0.001$).

## The relationship between the serum ALT levels and each component of MS in participants whose ALT levels were within the normal range

Table 3 shows the relationship between the serum ALT levels and each component of MS in participants whose ALT levels were within the normal range. In male participants, the serum ALT levels were significantly associated with waist circumference ($r = 0.321$, $p < 0.001$), SBP ($r = 0.080$, $p < 0.001$), DBP ($r = 0.184$, $p < 0.001$), FBS ($r = 0.094$, $p < 0.001$), and triglyceride ($r = 0.234$, $p < 0.001$). In female participants, the serum ALT levels were also significantly associated with waist circumference ($r = 0.343$, $p < 0.001$), SBP ($r = 0.227$, $p < 0.001$), DBP ($r = 0.185$, $p < 0.001$), FBS ($r = 0.195$, $p < 0.001$), and triglyceride ($r = 0.269$, $p < 0.001$). An

**Table 3. Correlations between serum ALT and each components of metabolic syndrome.**

| | Serum ALT level (U/L) | | | |
|---|---|---|---|---|
| | Male | | Female | |
| Variable | r | P | r | P |
| Waist circumference (cm) | 0.321 | <0.001 | 0.343 | <0.001 |
| SBP (mmHg) | 0.080 | <0.001 | 0.227 | <0.001 |
| DBP (mmHg) | 0.184 | <0.001 | 0.185 | <0.001 |
| FBS (mg/dL) | 0.094 | <0.001 | 0.195 | <0.001 |
| Triglyceride (mg/dL) | 0.234 | <0.001 | 0.269 | <0.001 |
| HDL cholesterol (mg/dL) | -0.098 | <0.001 | -0.140 | <0.001 |

*Abbreviations: ALT, alanine aminotransferase; SBP, systolic blood pressure; DBP, diastolic blood pressure; FBS, fasting blood sugar; HDL, high density lipoprotein.

'r' means the correlation coefficient.

**Table 4. Risk of metabolic syndrome in patients whose ALT levels were within the normal range.**

| Risk of metabolic syndrome | Male | | | | | Female | | | | |
|---|---|---|---|---|---|---|---|---|---|---|
| | <15 U/L | 15~30 U/L | | 30~40 U/L | | <10 U/L | 10~20 U/L | | 20~40 U/L | |
| | OR | OR | P value | OR | P value | OR | OR | P value | OR | P value |
| Unadjusted | Reference | 2.33 (2.10–2.56) | <0.001 | 4.57 (4.04–5.16) | <0.001 | Reference | 3.89 (3.38–4.47) | <0.001 | 10.37 (9.00–11.97) | <0.001 |
| Model 1 | Reference | 2.42 (2.18–2.69) | <0.001 | 5.31 (4.68–6.02) | <0.001 | Reference | 2.20 (1.90–2.55) | <0.001 | 5.28 (4.53–6.15) | <0.001 |
| Model 2 | Reference | 2.37 (2.13–2.64) | <0.001 | 5.12 (4.51–5.81) | <0.001 | Reference | 2.20 (1.89–2.55) | <0.001 | 5.27 (4.52–6.15) | <0.001 |
| Model 3 | Reference | 1.55 (1.38–1.74) | <0.001 | 2.48 (2.16–2.85) | <0.001 | Reference | 1.58 (1.35–1.86) | <0.001 | 2.67 (2.26–3.15) | <0.001 |

Abbreviations: ALT, alanine aminotransferase; OR, odds ratio.

Adjusted ORs were calculated in models 1, 2, and 3. The variables for adjustment were age in model 1, age and alcohol consumption in model 2, and age, alcohol consumption, and body mass index in model 3.

inverse correlation between HDL cholesterol and ALT activity was noted in both male ($r$ = -0.098, $p$ < 0.001) and female ($r$ = -0.140, $p$ < 0.001) participants.

## Risk of MS in participants whose ALT levels were within the normal range

Table 4 presents the risk of MS in participants whose ALT levels were within the normal range. We assessed the risk of MS with or without adjustment for the clinical variables: unadjusted and models 1 (age), 2 (age and alcohol consumption), and 3 (age, alcohol consumption, and BMI).

In male participants, the risk of MS increased in the ALT level ranges of 15–30 and 30–40 U/L compared with that in ALT<15 U/L, respectively, in the unadjusted model (OR, 2.33 [95% confidence interval {CI}, 2.10–2.56]; OR, 4.57 [95% CI, 4.04–5.16]). It also increased after adjustment for the clinical variables. In model 1, the risk of MS in these participants increased in the ALT level ranges of 15–30 and 30–40 U/L compared with that in ALT <15 U/L, respectively (aOR, 2.42 [95% CI, 2.18–2.69]; aOR, 5.31 [95% CI, 4.68–6.02]). In model 2, it increased in the ALT level ranges of 15–30 and 30–40 U/L compared with that in ALT <15 U/L, respectively (aOR, 2.37 [95% CI, 2.13–2.64]; aOR, 5.12 [95% CI, 4.51–5.81]). Moreover, in model 3, it increased in the ALT level ranges of 15–30 and 30–40 U/L compared with that in ALT <15 U/L, respectively (aOR, 1.55 [95% CI, 1.38–1.74]; aOR, 2.48 [95% CI, 2.16–2.85]).

In female participants, the risk of MS increased in the ALT level ranges of 10–20 and 20–40 U/L compared with that in ALT <10 U/L, respectively, in the unadjusted model (OR, 3.89 [95% CI, 3.38–4.47]; OR, 10.37 [95% CI, 9.00–11.97]). It also increased after adjustment for the clinical variables. In model 1, the risk of MS increased in the ALT level ranges of 10–20 and 20–40 U/L compared with that in ALT <10 U/L, respectively (aOR, 2.20 [95% CI, 1.90–2.55]; aOR, 5.28 [95% CI, 4.53–6.15]). In model 2, it increased in the ALT level ranges of 10–20 and 20–40 U/L compared with that in ALT <10 U/L, respectively (aOR, 2.20 [95% CI, 1.89–2.55; aOR, 5.27 [95% CI, 4.52–6.15]). Moreover, in model 3, it increased in the ALT level ranges of 10–20 and 20–40 U/L compared with that in ALT <10 U/L, respectively (aOR, 1.58 [95% CI, 1.35–1.86]; aOR, 2.67 [95% CI, 2.26–3.15]).

## Comparison of the risk of MS between the participants whose ALT levels were within and above the normal range

S1 and S2 Tables show the clinical characteristics of the participants with increased ALT levels (>40 U/L). In male participants, the prevalence rates of MS were 12.6%, 25.2%, 39.7%, and 50.8% in the ALT levels of <15, 15–30, 30–40, and >40 U/L, respectively. A dose-dependent

relationship between MS and ALT levels was noted in these participants ($p < 0.001$). In female participants, the prevalence rates of MS were 7.2%, 23.3%, 44.7%, and 57.8% in the ALT levels of <10, 10–20, 20–40, and >40 U/L, respectively. A dose-dependent relationship between MS and ALT levels was noted in these participants ($p < 0.001$). Then, we compared the risk of MS between the participants whose ALT levels were within and above the normal range (S3 Table). In male participants, the risk of MS increased in the ALT level ranges of 15–30, 30–40, and >40 U/L compared with that in ALT < 15 U/L, respectively, after adjustment for age, alcohol consumption, and BMI (aOR, 1.57 [95% CI, 1.40–1.76]; aOR, 2.54 [95% CI, 2.21–2.92]; and aOR, 3.63 [95% CI, 3.14–4.19], respectively). In female participants, the risk of MS increased in the ALT level ranges of 10–20, 20–40, and >40 U/L compared with that in ALT <10 U/L, respectively, after adjustment for age, alcohol consumption, and BMI (aOR, 1.59 [95% CI, 1.36–1.87]; aOR, 2.69 [95% CI, 2.28–3.18]; and aOR, 4.28 [95% CI, 3.46–5.31], respectively).

## Discussion

The present study investigated the risk of MS in participants without chronic liver disease and whose ALT levels were within the normal range (≤40 U/L) in the Korean general population. The prevalence of MS significantly increased as the ALT levels increased. There was an ALT-dependent relationship in the risk of MS in participants with normal ALT levels after adjustment for age, alcohol intake, and BMI.

The measurement of ALT, one of the many liver enzymes, is a widely used test for liver injury and applied in several clinical situations (acute and chronic liver disease, health checkup, preoperative examination, etc.). The ALT levels provide a basic clue to determine the presence of liver diseases including viral hepatitis, alcoholic liver disease, drug-induced liver injury, and NAFLD. Although ALT is perceived mainly as a liver enzyme, their increased levels are also associated with cardiovascular as well as liver diseases. In addition, such levels predict the development of type 2 diabetes mellitus and coronary heart disease [20,21] and are related to hypoxia in patients with obstructive sleep apnea [22] and associated with intracerebral hemorrhage [23].

MS is a cluster of clinical and laboratory findings consisting of high BMI, blood pressure, and glucose and triglyceride levels and low HDL level [24]. A meta-analysis showed that participants with MS have a twofold increased risk of cardiovascular disease, myocardial infarction, stroke, and cardiovascular disease-related mortality compared with those without MS [25]. MS is associated with increased ALT levels [8,9]. An Australian population-based cohort study showed that the ALT levels were strongly associated with the prevalence of MS [26]. A population-based Hispanic cohort study revealed that high prevalence rates of MS were observed in participants with increased ALT levels and male ones had increased ALT levels compared with the females [27]. A Chinese population-based cohort study demonstrated that the longitudinal increments of the ALT levels were related to an increased incidence of MS [28]. MS is related to an elevated ALT level in patients with diabetes mellitus [29]. Also, it is well known that MS is closely related to NAFLD. The prevalence of MS in patients with NAFLD increases as the BMI increases [30]. Furthermore, the increased ALT levels are significantly associated with MS in patients with NAFLD and viral hepatitis [31,32].

This study investigated the risk of MS in participants whose ALT levels were within the normal range (≤40 U/L). The prevalence rates of MS were 24.6% and 26.3% in male and female participants, respectively. Also, the proportions of MS in male and female participants whose ALT levels were within the ULN (male, 30~40 U/L; female, 20~40 U/L) were 39.7% and 44.7%, respectively. The prevalence rate of MS was approximately 40% in the Chinese general

population with an ALT level within the normal range (≤40 U/L) [33]. The ALT level within the normal range (≤40 U/L) is closely correlated with the severity of MS in a population-based cohort in Germany [34]. The risk of MS is associated with increased ALT levels within the normal range (≤43 U/L) in the Korean population study [35]. Arterial stiffness and MS were related to the ALT levels within the normal range (≤40 U/L) regardless of alcoholic consumption [36]. The present study also examined the relationship between the score of MS components and ALT levels within the normal range (≤40 U/L). The ALT level significantly increased as the score of MS components increased. Considering the results of the aforementioned and present study, the risk of MS exists even in participants whose ALT levels were within the normal range. Moreover, there is an ALT-dependent relationship in the risk of MS in participants with normal ALT level. Similarly, the risk of liver disease can still exist even if the ALT level is within the normal range. The risk of disease progression is also still noted in patients with chronic hepatitis B or C despite having ALT levels within the normal range [10,11]. Also, the majority of patients with NAFLD (~80%) have ALT levels within the normal range levels [37]. Amarapurkar *et al.* reported that an abnormal ALT level is observed in 16% of patients with NAFLD diagnosed by ultrasonography [38].

This study has several limitations. Firstly, imaging study or liver biopsy was not checked. Therefore, this study has limitations in detailed information for liver diseases such as hepatic steatosis or liver cirrhosis. Secondly, the relationship between MS and the ALT level change was not evaluated because the present study was conducted based on a cross-sectional study design. Finally, the association between the serum ALT levels and inflammatory markers such as tumor necrosis factor-alpha, plasminogen activator inhibitor 1, interleukin-6, leptin, and adiponectin was not elucidated. In spite of these limitations, this study investigated the relationship between MS and a normal ALT level in a large-scale general population.

## Conclusions

In conclusion, the risk of MS significantly increases as the ALT level increases within the normal ALT level. Even if the ALT level is within the normal range, it may be a useful marker to consider the presence of MS.

## Supporting information

**S1 Table. Baseline characteristics of the participants.**
(DOCX)

**S2 Table. Clinical characteristics of the participants according to ALT level.**
(DOCX)

**S3 Table. Risk of metabolic syndrome according to serum ALT level.**
(DOCX)

## Author Contributions

**Conceptualization:** Ju-Yeon Cho, Jae Yoon Jeong, Won Sohn.

**Data curation:** Jae Yoon Jeong, Won Sohn.

**Formal analysis:** Won Sohn.

**Funding acquisition:** Ju-Yeon Cho.

**Investigation:** Ju-Yeon Cho, Won Sohn.

**Methodology:** Ju-Yeon Cho, Won Sohn.

**Supervision:** Won Sohn.

**Writing – original draft:** Ju-Yeon Cho, Jae Yoon Jeong, Won Sohn.

**Writing – review & editing:** Ju-Yeon Cho, Jae Yoon Jeong, Won Sohn.

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
