## [Decision Letter · Decision Letter 0]

24 Jan 2020

PONE-D-19-34865

Risk of metabolic syndrome in subjects with a normal range of alanine aminotransferase: a population-based nationwide study

PLOS ONE

Dear Dr Sohn,

Thank you for submitting your manuscript to PLOS ONE. After careful consideration, we feel that it has merit but does not fully meet PLOS ONE’s publication criteria as it currently stands. Therefore, we invite you to submit a revised version of the manuscript that addresses the points raised during the review process.

We would appreciate receiving your revised manuscript by Mar 09 2020 11:59PM. To enhance the reproducibility of your results, we recommend that if applicable you deposit your laboratory protocols in protocols.io, where a protocol can be assigned its own identifier (DOI) such that it can be cited independently in the future. For instructions see: http://journals.plos.org/plosone/s/submission-guidelines#loc-laboratory-protocols

We look forward to receiving your revised manuscript.

Kind regards,

Ming-Lung Yu, MD, PhD

Academic Editor

PLOS ONE

Journal Requirements:

2. Thank you for stating the following on the Title page of your manuscript:

"This study was supported by research fund from Chosun University, 2016. The fund had no role in study design, data collection and analysis, decision to publish,

or preparation of the manuscript."

Reviewers' comments:

Reviewer's Responses to Questions

**Comments to the Author**

1. Is the manuscript technically sound, and do the data support the conclusions?

Reviewer #1: Yes

Reviewer #2: Yes

2. Has the statistical analysis been performed appropriately and rigorously? 

Reviewer #1: Yes

Reviewer #2: Yes

3. Have the authors made all data underlying the findings in their manuscript fully available?

Reviewer #1: Yes

Reviewer #2: Yes

4. Is the manuscript presented in an intelligible fashion and written in standard English?

Reviewer #1: Yes

Reviewer #2: No

5. Review Comments to the Author

Reviewer #1: The community-based study included 43402 adults with ALT<=40 U/L to investigate the associations between serum ALT levels and metabolic syndrome (MS). They found that the prevalence of MS increased with serum ALT levels. In addition, the probability of MS increased with serum ALT levels with dose-response trends. The authors concluded that ALT could be used as a marker to evaluate MS, particularly in a general population with normal ALT levels.

The work is clear and straightforward. However, there are still some comments:

1. Although the work is interesting, the clinical utility is limited. There is already clear definitions for MS as the authors indicated in the methods (MS was defined by the updated National Cholesterol Education Program Adult Treatment Panel III standards). Are there any benefits by considering ALT? By including ALT, is it superior to the original definition?

2. It will be interesting to know whether serum ALT levels was correlated to any inflammatory markers or not.

3. It is suggested to calculate fatty liver index and examine the associations between ALT and FLI.

Reviewer #2: The authors aimed to investigate the risk of metabolic syndrome (MS) in subjects with normal ALT level in the general population. They concluded that despite within the normal range of ALT, the risk of MS increased as the ALT levels increased. In general, the topic of this work is interesting. However, some issues should be further reconsidered or corrected.

1. The authors should provide the complete data of metabolic syndrome including systolic and diastolic blood pressure. Also, the correlation of each component of metabolic syndrome with serum ALT level should be evaluated.

2. The subjects with elevated ALT levels should not be excluded and could serve as positive controls.

3. The information of medication history including anti-diabetics, statin and anti-hypertensive drugs should be provided.

4. It is noted that the manuscript is filled with some grammatical errors that need to be corrected.

6. PLOS authors have the option to publish the peer review history of their article (what does this mean?). If published, this will include your full peer review and any attached files.

Reviewer #1: No

Reviewer #2: No

---

## [Author Response · Author response to Decision Letter 0]

20 Feb 2020

Reviewer #1: The community-based study included 43402 adults with ALT<=40 U/L to investigate the associations between serum ALT levels and metabolic syndrome (MS). They found that the prevalence of MS increased with serum ALT levels. In addition, the probability of MS increased with serum ALT levels with dose-response trends. The authors concluded that ALT could be used as a marker to evaluate MS, particularly in a general population with normal ALT levels.

The work is clear and straightforward. However, there are still some comments:

1. Although the work is interesting, the clinical utility is limited. There is already clear definitions for MS as the authors indicated in the methods (MS was defined by the updated National Cholesterol Education Program Adult Treatment Panel III standards). Are there any benefits by considering ALT? By including ALT, is it superior to the original definition?

: We appreciate your valuable comment. The aim of the study was to investigate the association between serum ALT and MS. The results of our study cannot imply that the addition of ALT in the diagnostic criteria of MS would be superior to the current standards. However, ALT is a basic laboratory test done frequently in the daily clinic and increases in ALT can be a useful marker that implies the presence of MS. 

2. It will be interesting to know whether serum ALT levels was correlated to any inflammatory markers or not. 

: Thank you for your comment. Unfortunately, the present study did not check inflammatory markers such as tumor necrosis factor-alpha (TNF-α), plasminogen activator inhibitor 1 (PAI-1), interleukin-6 (IL-6), leptin or adiponectin. We added this limitation in the discussion section of the revised manuscript (page 14).

Finally, the present study did not clarify the association between serum ALT level and the inflammatory markers such as tumor necrosis factor-alpha (TNF-α), plasminogen activator inhibitor 1 (PAI-1), interleukin-6 (IL-6), leptin and adiponectin.

3. It is suggested to calculate fatty liver index and examine the associations between ALT and FLI.

: We appreciate your valuable comment. As you commented, we newly analyzed the relationship between serum ALT and fatty liver. However, we could not calculate the fatty liver index (FLI) because our data did not include γ-glutamyl-transferase. Fatty liver index consists of triglyceride, BMI, γ-glutamyl-transferase and waist circumference. Therefore, we calculated the hepatic steatosis index (HSI) instead of the fatty liver index (FLI). HSI consists of ALT, AST, BMI, diabetes, and sex (Dig Liver Dis 2010;42:503-8.). We analyzed the association between ALT level and HSI. There was a dose-dependent increase of HSI as ALT levels increased in both male and female subjects (Table 2). In male subjects, HSI was 28.0 ± 3.2, 31.9 ± 3.8 and 35.9 ± 4.3 in ALT 0-15 U/L, 15-30 U/L, and 30-40 U/L, respectively (p<0.001). In female subjects, HSI was 28.0 ± 3.0, 31.4 ± 3.8 and 35.7 ± 4.4 in ALT <10 U/L, 10-20 U/L, 20-40 U/L and >40 U/L, respectively (p<0.001). We added these findings in the method section and results section of the revised manuscript (page 7 & 8 and Table 2).

“Hepatic steatosis index (HSI) was calculated to assess the relationship between serum ALT level and fatty liver.” (page 7)

“The mean level of HSI in male and female subjects was 31.6 ± 4.5 and 32.0 ± 4.5, respectively.” (page 8)

There were dose-dependent increases of BMI, waist circumference, SBP, DBP, fasting blood glucose (FBS), total cholesterol, triglyceride, HSI, the prevalence of diabetes mellitus, and the prevalence of MS as ALT level increased in both male and female subjects.

Reviewer #2: The authors aimed to investigate the risk of metabolic syndrome (MS) in subjects with normal ALT level in the general population. They concluded that despite within the normal range of ALT, the risk of MS increased as the ALT levels increased. In general, the topic of this work is interesting. However, some issues should be further reconsidered or corrected.

1. The authors should provide the complete data of metabolic syndrome including systolic and diastolic blood pressure. Also, the correlation of each component of metabolic syndrome with serum ALT level should be evaluated.

: According to your comments, we added the data for systolic and diastolic blood pressure in the results section and Table 1 & 2 of the revised manuscript (page 8)

“Systolic blood pressure (SBP) of male and female subjects was 121.2 ± 15.9 and 116.4 ± 18.0, respectively. Diastolic blood pressure (DBP) of male and female subjects was 78.1 ± 10.6 and 73.6 ± 10.0, respectively.”

Also, we analyzed the correlation of each component of metabolic syndrome with the serum ALT level (Table 4). We added these results in table 4 of the revised manuscript (page 9-10).

“The relationship between serum ALT level and each component of MS in subjects with a normal range of ALT are shown in Table 4. In male subjects, serum ALT level were significantly associated with waist circumference (r=0.321, p<0.001), SBP (r=0.080, p<0.001), DBP (r=0.184, p<0.001), FBS (r=0.094, p<0.001), and triglyceride (r=0.234, p<0.001). In female subjects, serum ALT level were significantly associated with waist circumference (r=0.343, p<0.001), SBP (r=0.227, p<0.001), DBP (r=0.185, p<0.001), FBS (r=0.195, p<0.001), and triglyceride (r=0.269, p<0.001). There was an inverse correlation between HDL cholesterol and ALT activity in both male (r=-0.098, p<0.001) and female (r=-0.140, p<0.001).”

2. The subjects with elevated ALT levels should not be excluded and could serve as positive controls.

: According to your comments, we added the data for the subjects with elevated ALT levels in the method section, results section, and supplementary document of the revised manuscript (page 6). 

“Finally, 43,402 subjects within the normal range of alanine aminotransferase (ALT ≤40 U/L) were included in the final analysis and we compared these subjects to 3,426 subjects with ALT level >40 U/L (Figure 1).” 

We also compared the risk of metabolic syndrome the subjects with normal range and abnormal range of ALT (page 11-12).

“Clinical characteristics of the subjects with an abnormal range of ALT (>40 U/L) are presented in Supplementary table 1 and 2. In male subjects, the prevalence of MS was 12.6%, 25.2%, 39.7% and 50.8% in ALT <15 U/L, 15-30 U/L, 30-40 U/L and >40 U/L, respectively. There was a dose-dependent relationship between MS and ALT level in male subjects (p<0.001). In female subjects, the prevalence of MS was 7.2%, 23.3%, 44.7% and 57.8% in ALT <10 U/L, 10-20 U/L, 20-40 U/L and >40 U/L, respectively. There was a dose-dependent relationship between MS and ALT level in female subjects (p<0.001). We compared the risk of MS between the subjects with normal range and abnormal range of ALT (Supplementary table 3). The risk of MS in male subjects is increased when in ALT 15-30 U/L, 30-40 U/L, and >40 U/L compared to ALT<15 U/L, respectively in adjusting age, alcohol consumption, and BMI (aOR 1.57 with 95% CI: 1.40-1.76, aOR 2.54 with 95% CI: 2.21-2.92 and aOR 3.63 with 95% CI: 3.14-4.19, respectively). The risk of MS in female subjects is increased when in ALT 10-20 U/L, 20-40 U/L, and >40 U/L compared to ALT<10 U/L, respectively in adjusting age, alcohol consumption, and BMI (aOR 1.59 with 95% CI: 1.36-1.87, aOR 2.69 with 95% CI: 2.28-3.18 and aOR 4.28 with 95% CI: 3.46-5.31, respectively).” 

3. The information of medication history including anti-diabetics, statin and anti-hypertensive drugs should be provided.

: As you commented, we added the information regarding the use of anti-diabetic agents, lipid lowering agents, and anti-hypertensive agents. The proportion of the usage of anti-diabetic agents in male and female subjects was 7.7% and 6.0%, respectively. The proportion of the usage of lipid lowering agents in male and female subjects was 4.7% and 5.6%, respectively. The proportion of the usage of anti-hypertensive agents in male and female subjects was 18.8% and 18.8%, respectively. We added these results in Table 1 of the revised manuscript.

4. It is noted that the manuscript is filled with some grammatical errors that need to be corrected.

: As you have commented, the manuscript has been edited for grammatical errors.

---

## [Decision Letter · Decision Letter 1]

17 Mar 2020

PONE-D-19-34865R1

Risk of metabolic syndrome in subjects within the normal range of alanine aminotransferase: a population-based nationwide study

PLOS ONE

Dear Dr Sohn,

Thank you for submitting your manuscript to PLOS ONE. After careful consideration, we feel that it has merit but does not fully meet PLOS ONE’s publication criteria as it currently stands. Therefore, we invite you to submit a revised version of the manuscript that addresses the points raised during the review process.

The English should be revised by native English speakers.  

We would appreciate receiving your revised manuscript by May 01 2020 11:59PM. To enhance the reproducibility of your results, we recommend that if applicable you deposit your laboratory protocols in protocols.io, where a protocol can be assigned its own identifier (DOI) such that it can be cited independently in the future. For instructions see: http://journals.plos.org/plosone/s/submission-guidelines#loc-laboratory-protocols

We look forward to receiving your revised manuscript.

Kind regards,

Ming-Lung Yu, MD, PhD

Academic Editor

PLOS ONE

Reviewers' comments:

Reviewer's Responses to Questions

**Comments to the Author**

1. If the authors have adequately addressed your comments raised in a previous round of review and you feel that this manuscript is now acceptable for publication, you may indicate that here to bypass the “Comments to the Author” section, enter your conflict of interest statement in the “Confidential to Editor” section, and submit your "Accept" recommendation.

Reviewer #1: All comments have been addressed

Reviewer #2: All comments have been addressed

2. Is the manuscript technically sound, and do the data support the conclusions?

Reviewer #1: Yes

Reviewer #2: Yes

3. Has the statistical analysis been performed appropriately and rigorously? 

Reviewer #1: Yes

Reviewer #2: Yes

4. Have the authors made all data underlying the findings in their manuscript fully available?

Reviewer #1: Yes

Reviewer #2: Yes

5. Is the manuscript presented in an intelligible fashion and written in standard English?

Reviewer #1: Yes

Reviewer #2: Yes

6. Review Comments to the Author

Reviewer #1: all comments have been addressed adequately. It is suggested that the article could be polished by an English editor before formal publication.

Reviewer #2: The authors aimed to investigate the risk of metabolic syndrome (MS) in subjects with normal ALT level in the general population. In general, this is an interesting topic and a clearly written paper. Also, the authors have well responded to the reviewer’s suggestions.

7. PLOS authors have the option to publish the peer review history of their article (what does this mean?). If published, this will include your full peer review and any attached files.

Reviewer #1: No

Reviewer #2: No

---

## [Author Response · Author response to Decision Letter 1]

23 Mar 2020

Editor’s comment: The English should be revised by native English speakers.

: We appreciate your valuable comment. According to your comment, the revised manuscript was examined by an English editing service (www.editage.co.kr).

Reviewer #1: all comments have been addressed adequately. It is suggested that the article could be polished by an English editor before formal publication.

: We appreciate your valuable comment. According to your comment, the revised manuscript was examined by an English editing service (www.editage.co.kr).

Reviewer #2: The authors aimed to investigate the risk of metabolic syndrome (MS) in subjects with normal ALT level in the general population. In general, this is an interesting topic and a clearly written paper. Also, the authors have well responded to the reviewer’s suggestions.

: We appreciate your valuable comment.

---

## [Editor Report · Decision Letter 2]

25 Mar 2020

Risk of metabolic syndrome in participants within the normal range of alanine aminotransferase: a population-based nationwide study

PONE-D-19-34865R2

Dear Dr. Sohn,

We are pleased to inform you that your manuscript has been judged scientifically suitable for publication and will be formally accepted for publication once it complies with all outstanding technical requirements.

With kind regards,

Ming-Lung Yu, MD, PhD

Academic Editor

PLOS ONE
---

## [Editor Report · Acceptance letter]

27 Mar 2020

PONE-D-19-34865R2 

Risk of metabolic syndrome in participants within the normal range of alanine aminotransferase: a population-based nationwide study 

Dear Dr. Sohn:

I am pleased to inform you that your manuscript has been deemed suitable for publication in PLOS ONE. Congratulations! Your manuscript is now with our production department. 

With kind regards,

on behalf of

Dr. Ming-Lung Yu 

Academic Editor

PLOS ONE